# Cardiovascular Risk Factors and Family History of Major Thrombotic Events in Children with Migraine: A 12-Year Retrospective Single-Centre Study

**DOI:** 10.3390/jcm12072582

**Published:** 2023-03-29

**Authors:** Roberta Rossi, Stefania Benetti, Barbara Lauria, Giulia Grasso, Emanuele Castagno, Fulvio Ricceri, Claudia Bondone, Antonia Versace

**Affiliations:** 1Department of Pediatric Emergency, Pediatric Headache Centre, Regina Margherita Children’s Hospital, A.O.U. Città della Salute e della Scienza di Torino, Piazza Polonia 94, 10126 Turin, Italy; 2Department of Pediatrics, Ospedale degli Infermi, ASL TO3, Via Rivalta 29, 10098 Rivoli, Italy; 3Department of Clinical and Biological Sciences, University of Turin, Regione Gonzole 10, 10043 Orbassano, Italy

**Keywords:** children, migraine, cardiovascular events, International Classification of Headache Disorders, third edition (ICHD-III)

## Abstract

Background: Migraine is one of the most frequent primary headaches in childhood. The role of thrombotic predisposition in its pathogenesis is debated. Our aim was to analyse the cardiovascular risk factors and family history of major thrombotic events in children with migraine. Methods: A retrospective, single-centre study was performed over 12 years. Our headache centre record database was screened for migraine with aura (MA) and migraine without aura (MO) on the basis of the ICHD-II (until 2013) and III criteria. A control group of otherwise healthy children was recruited. Descriptive and multivariate analyses are provided; significance was set at *p* < 0.05. Results: Migraine was diagnosed in 930 children (24.7% MA); 73.3% were 9–14 years old. Children with MA were older (*p* < 0.001). A family history of cerebral ischemic events at ≤50 years old was more commonly reported by children with MA than those with MO (*p* < 0.001) and those in the control group (*p* = 0.001). Children with MA showed a higher risk of a family history of cerebral ischemic events at ≤50 years old than children with MO (OR: 2.6) and those in the control group (OR: 3.1). When comparing the family history of DVT, we observed a significantly increased risk for MA vs. MO (OR: 2.9). Conclusion: A family history of cerebral ischemic events at ≤50 years old leads to an increased risk of MA. Further studies are needed to explore such an association.

## 1. Introduction

Migraine is one of the most common types of primary headaches in childhood and adolescence and represents the leading cause of disability in the working population (15–49-year-olds) worldwide, with females affected nearly three times more often than males [1]. It negatively affects quality of life in terms of daily activities, school performance, social relationships, and use of medication [1,2]. The estimated prevalence of migraine in childhood is 9.1%; it is similar in males and females at younger ages but rapidly increases in females during adolescence [3]. Some authors have shown that migraine persists in adulthood in 20–48% of adolescents and shifts to other types of headaches in 11–37% of cases, with a higher risk in children who have first-degree migraineur relatives [4,5]. Furthermore, there is evidence to suggest that early diagnosis and appropriate treatment during childhood and adolescence may prevent the evolution of chronic disease, thus reducing disability and improving long-term outcomes and prognosis [6].

An association between migraine, a pro-thrombotic state, and thrombotic events (such as stroke and myocardial infarction) has been proposed in adults [7,8,9]. The mechanisms underlying the association between migraine and thrombotic events are not fully understood, but several factors may be involved. Migraine is associated with classic thrombotic risk factors, such as hypertension, hyperlipidaemia, insulin resistance, obesity, and smoking. Indeed, some authors have proposed migraine as an independent thrombotic risk factor itself [7,8,9,10,11,12]. Such an association has also been investigated in children. A large retrospective study showed an increased risk of ischemic stroke in adolescents with migraine and suggested a hormonal effect to explain it [13].

The aim of our study was to analyse the possible association between migraine, cardiovascular risk factors, and a family history of major thrombotic events in a cohort of children evaluated at our headache centre.

## 2. Materials and Methods

This was a retrospective, case–control, observational study on all children ≤18 years old screened for episodic migraine with aura (MA) or migraine without aura (MO) at our tertiary Pediatric Headache Centre over 12 years, between January 2006 and September 2018. Migraine was diagnosed according to the International Headache Society’s ICHD, second edition (ICHD-II), until 2013 [14] and according to ICHD-III thereafter [15].

Exclusion criteria were as follows: any disorder related to headache (intracranial tumours, inflammatory diseases, nerve trunk pain, any disorder of facial cranial structures, cranial neuralgias, or recent head injuries), chronic daily headaches, cluster headaches, chronic migraines, any form of headache or migraine other than episodic MO or MA, and chronic systemic diseases. Patients with insufficient available data were also excluded.

The medical record database of our centre was screened for information on the demographic profile of children (age, sex, and ethnicity), clinical features of migraine, associated symptoms, and personal and family medical history. Headache diaries were used to evaluate the duration, severity, and intensity of pain.

Detailed personal medical history was investigated, with particular attention to cardiovascular risk factors: hypertension, overweight/obesity, diabetes, oral contraceptives, smoking, and intake of alcohol. The family history (parents, grandparents, uncles, and aunts) of idiopathic headache and major thrombotic events at ≤50 years old (stroke, transitory ischemic attack (TIA), myocardial infarction, deep venous thrombosis (DVT), and pulmonary embolism) was recorded. All the information was obtained from interviews with parents and by consulting relevant medical reports. All the enrolled participants underwent physical and neurological evaluations during their initial visit to our centre.

A control group of otherwise healthy children without a personal history of headaches was recruited from patients who were admitted to the Department of Pediatric Emergency of our children’s hospital for acute respiratory or gastrointestinal diseases between 2006 and 2018. The control group was screened for information relative to the demographic profile of children (age, sex, and ethnicity) and family history of major thrombotic events at ≤50 years old.

The study was performed according to the international regulatory guidelines and current codes of Good Epidemiological Practices. According to European regulations, Italian observational studies using data obtained without any additional therapy or monitoring procedures do not need the approval of an independent ethical committee.

### Statistical Analysis

Statistical analysis was performed using SAS software for Windows, version 9.2 (SAS Institute Inc., Cary, NC, USA). Data are described as the mean and standard deviation (SD) or the median and interquartile range (IQR) for continuous variables (as appropriate) and the absolute and relative frequencies for categorical variables. A Chi-squared test or Fisher’s exact test was used to evaluate proportions, as appropriate. The Mann–Whitney U test was performed to evaluate comparisons within non-parametric quantitative variables. Multivariate logistic models were built to evaluate the relationships between migraine (MA and MO) and both personal and clinical variables. Odds ratios (OR) were calculated with the relative confidence intervals (CI). Individuals with missing values for at least one of the variables in the model were excluded from the analysis.

Statistical significance was set at *p* < 0.05; all *p*-values were based on two-tailed tests.

## 3. Results

### 3.1. Epidemiological Features

The study population consisted of 930 children (mean age: 10.3 ± 2.6 years): 230 patients (24.7%) suffered from MA, while 700 (75.3%) had MO. Overall, 484 patients were male (52.0%), but we observed a significant prevalence of males among patients with MO (395/700; 56.4%) and females among patients with MA (141/230; 61.3%) (*p* < 0.001). The control group consisted of 300 otherwise healthy children without headaches (mean age: 8.5 ± 4.3 years); 152 (50.7%) were male. All the demographic characteristics of the study population and the control group are reported in Table 1.

After stratification into five age groups (≤5 years, 6–8 years, 9–11 years, 12–14 years, and 15–18 years), we observed that the majority of migraine diagnoses were in children aged 9–14 years (73.3%) (Table 2). The majority of MA diagnoses were in children aged between 12 and 14 years (50.9%), while the majority of MO diagnoses were in children aged between 9 and 11 years (46.9%). Children with MA were significantly older than those with MO (mean age: 12 ± 2.3 and 9.8 ± 2.4, respectively) (*p* < 0.001). Males with MO prevailed significantly in the 9–11-year age group (*p* = 0.002), while females with MA prevailed significantly in the 12–14-year age group (*p* = 0.001) (Table 2).

Physical and neurological examinations at the initial visit were negative for all patients in the study group, and the clinical presentation of migraine was unchanged between the first appearance and the time of consultation. Among children with MA, 179 showed visual aura (77.8%), 76 showed sensorial aura (33.0%), 44 showed motor aura (19.1%), and 43 showed dysarthria (18.7%); overall, 91 children (39.6%) showed two or more aura features in each episode.

### 3.2. Cardiovascular Risk Factors of the Study Population

A significant difference between children with MO and MA was found only for obesity and overweight. In particular, 17 children (7.3%) with MA and 29 (4.3%) with MO were obese or overweight (*p* = 0.02). Hypertension was diagnosed in three children (1.4%) with MA and six (0.9%) with MO (*p* = 0.23) One child (0.4%) with MA and four (0.6%) with MO had diabetes (*p* = 0.58). Only one child (0.15%) with MO was taking oral contraceptives. No patients reported any intake of alcohol, and only one child with MA (0.4%) reported smoking. None of the patients in the entire sample had cerebrovascular disease.

### 3.3. Family History of Major Thrombotic Events

The global prevalence of family history of major thrombotic events at ≤50 years old was significantly higher in MA (20.3%) than MO (13.4%) (*p* = 0.005) and the control group (14.7 %) (*p* = 0.02) (Table 3). In detail, the prevalence of a family history of stroke and TIA was significantly higher in children with MA (12.4%) than in children with MO (5.1 %) (*p* < 0.001) and those in the control group (4.3%) (*p* < 0.001). The prevalence of a family history of DVT was significantly higher in children with MA (4.6%) than in those with MO (1.7 %) (*p* = 0.01), but no difference was found when comparing both the MA and MO groups with the control group (2.7%) (*p* = 0.09 and *p* = 0.11, respectively).

No significant difference was found for family history of myocardial infarction and pulmonary embolism between patients with MA, patients with MO, and controls.

When comparing the family history of stroke and TIA, we observed a moderately increased risk in the MA group vs. the MO group (OR = 2.6; CI 1.54–4.46) and vs. the control group (OR = 3.1; CI 1.6–6.2) (Table 3). Moreover, when comparing the family history of DVT, we observed a moderately increased risk in the MA group vs. the MO group (OR = 2.86; CI 1.2–6.8) but not in the MA group vs. the control group (OR = 1.76; CI 0.68–4.5) or the MO group vs. the control group (OR = 0.6; CI 0.24–1.5).

## 4. Discussion

This retrospective study provides new insights into the association between migraine and thrombotic risk in childhood and confirms the epidemiological features already reported in our previous study on patients with headaches in the emergency department [16].

The association between cardiovascular risk factors, family history of major thrombotic events at a young age, and pediatric migraine is debated. A recent study on 83 children with primary headaches showed no significant difference in blood levels of selected biomarkers of vascular changes indicative of atherosclerosis and that primary headaches are often related to a family history of cardiovascular disease [17].

On the other hand, migraine is recognised as a possible independent risk factor for ischemic stroke in young adults, and MA in particular is reported to double the risk of stroke [10,12]. However, the real mechanism underlying such an association is not fully understood. Several elements may be involved: the first is cortical spreading depression (a self-propagating wave of neuronal and glial depolarization that spreads across the cerebral cortex), but genetic, inflammatory, coagulation, and endothelial factors may also play a role [18]. The overlap between clinical and neuroimaging features and genetic substrates makes differential diagnosis between MA and ischemic episodes difficult, and this has led some authors to consider migrainous aura as a TIA equivalent [18]. Several studies have demonstrated the association between MA and white matter changes, subclinical infarct-like lesions, and volumetric change in the grey and white matter on imaging. These may be partially caused by alterations in resting cerebral blood flow and are likely directly associated with chronic long-standing migraine and frequency of attacks [18]. One hypothesis is the association between migraine and common risk factors for stroke, such as hypertension, hyperlipidaemia, obesity, smoking, cardioembolic diseases, cervical artery dissection, and hypercoagulability conditions [7,8,9,19,20,21]. Furthermore, migraine-associated genes expressed in vascular and smooth muscle have been shown to contribute to migraine pathogenesis, supporting shared polygenic risk between migraine, stroke, and thrombotic events [7,8,9,22,23,24]. Several genetic syndromes characterised by the occurrence of both migraine and cerebrovascular events strengthen the hypothesis of a genetic connection. The most well-known is cerebral autosomal-dominant arteriopathy with subcortical infarcts and leukoencephalopathy (CADASIL), sustained by mutations in NOTCH3. Another example is mitochondrial encephalomyopathy, lactic acidosis, and stroke-like episodes (MELAS), involving MT-TL1 mutations [18]. On the other hand, mutations in the four genes (CACNA-1A, ATP1A2, SCN1A, and PRRT2) identified in patients with familial hemiplegic migraine (FHM) are associated with a reduced threshold value of cortical spreading depression, which is also responsible for auras [25]. Moreover, an experimental study suggested that glutamatergic hyperexcitability associated with migraine mutations makes the brain more susceptible to ischemic depolarization [26]. As a result, the minimum critical level of blood flow required for tissue survival is elevated, and infarction occurs, even in mildly ischemic tissues. In addition, in our study, we observed that children with MA had a 2.6- and 3.1-fold higher risk of showing a family history of cerebral ischemic events at ≤50 years old compared with children in the MO and control groups, respectively. Moreover, children with MA had a similar increased risk of showing a family history of DVT. Our results are consistent with a previous case–control study on children and young adults, in which a family history of stroke was 2.1-fold more common among parents and grandparents of male patients with migraine [23]. The authors speculated that migraine and vascular disorders have common pathogenic mechanisms and that genetic susceptibility plays a role in increasing the risk of migraine in the offspring of families with one or more thrombotic conditions.

In accordance with the literature, our observations also suggest common pathogenic factors underlying MA and thrombotic disease. Therefore, when approaching children with MA, early detection of risk factors for thrombotic events is mandatory in order to improve prevention and therapy and reduce the overall cardiovascular risk. It is thus necessary to evaluate the presence of both anamnestic and hereditary risk factors, such as the family history of thrombotic events at ≤50 years old; high levels of lipoprotein A, protein S, and protein C deficiency; antithrombin III deficiency; and factor V Leiden G1691A and prothrombin G20210A mutations. Moreover, it is very important to promptly identify and fix acquired risk factors such as obesity, dyslipidemia, hypertension, smoking, and estrogen-containing oral contraceptives. In fact, although such risk factors often do not imply any therapeutic indication, it is important to intervene in lifestyle with adequate nutrition and appropriate physical activity.

Among cardiovascular risk factors, we observed a significantly higher prevalence of obesity and overweight in children with MA compared to those with MO.

Previous studies have shown that obesity is associated with an increased risk of migraine both in adults [27] and children [28]. The pathophysiology of obesity and migraine is multifactorial and involves both central and peripheral pathways [29]. Indeed, several hypothalamic peptides, proteins, and neurotransmitters involved in feeding regulation (including serotonin, orexin, and adipocytokines) probably contribute to the pathogenesis of migraine [29], along with lifestyle and environmental factors [20]. For these reasons, it is not easy to distinguish whether obesity itself may be a risk factor for migraine or whether some of the risk factors for obesity are also risk factors for migraine and vice versa. Moreover, pediatric research has confirmed a higher frequency and severity of migraine attacks among obese children than in normal-weight controls [23]. On the other hand, weight loss may contribute to reducing the incidence and severity of headaches and represents an important therapeutic strategy, while lifestyle and environmental factors (such as smoking and low physical activity) are associated with recurrent and chronic headaches in adolescents, both independently and in combination [20]. Unfortunately, our results do not allow us to speculate on the possible role of individual acquired factors, such as smoking, alcohol intake, and oral contraceptives. The analysis of a larger population, especially in the 15–18-year age group, might be helpful for this purpose.

In our sample, the majority of children were diagnosed with migraine between 9 and 14 years old; this is consistent with epidemiologic studies reporting a steady increase in migraine prevalence from preschool to school age, with a sharp increase in pubertal children and adolescents [30,31,32]. Moreover, children with MA were significantly older and more frequently female compared with those with MO, as already reported by other authors [16,31,33]. These aspects are crucial, as behavioural risk factors for migraine and thrombotic events, such as smoking and alcohol intake, may be acquired early at school age and may persist in adulthood [34]. Therefore, early preventive strategies should be put in place and implemented in this age group, either at healthcare facilities such as the headache centre or through dedicated programs in schools.

The association between MA and a family history of major thrombotic events at ≤50 years old allows the identification of patients who need early testing for both congenital and acquired thrombotic risk factors. Our study is important because it underlines the need to collect a detailed personal and family history of children with migraine, with particular attention to major thrombotic events at young ages. This is a rapid and simple practice that may effectively improve the patient’s outcomes through careful follow-up and early preventive and therapeutic strategies.

Our study also has some limitations. The retrospective nature of our single-centre research and the relatively small sample size, in particular regarding adolescents ≥15 years, could be responsible for the lack of information on personal risk factors, such as oral contraceptives, smoking, and alcohol intake. Furthermore, we did not obtain any information about personal risk factors from children in the control group.

## 5. Conclusions

In our study, we observed that children with MA had a higher risk of showing a family history of cerebral ischemic events and DVT.

Our findings support the hypothesis of a common etiopathology underlying MA and cardiovascular diseases. Therefore, the accurate detection of a family history of early ischemic events and any personal inborn or acquired risk factor is mandatory in children as well as adults in order to promptly apply adequate preventive or therapeutic strategies.

Further studies are needed to explore the mechanism underlying the potential association between MA and cardiovascular events and, in particular, to evaluate how inherited and acquired risk factors influence the natural history and management of MA. The identification of effective biomarkers to stratify the risk of major thrombotic events is desirable, even though such events are rare in children.

## 6. Take-Home Points

Children with migraine with aura showed had a higher risk of showing a family history of cerebral ischemic events and deep venous thrombosis;The detection of a family history of early ischemic events and any personal inborn or acquired risk factor is mandatory in children with migraine as well as in adults;The identification of effective biomarkers to stratify the risk of major thrombotic events in patients with migraine with aura is desirable.

## Figures and Tables

**Table 1 jcm-12-02582-t001:** Epidemiological features of the study population with migraine (1A) and the control group (1B).

**1A—Children with Migraine**	
**Variables**	Patients (*n* = 930)
**Sex, M/F—*n* (%)**	484/446 (52.0/48.0)
**Age at diagnosis—mean ± SD (years)**	10.3 ± 2.6
**Ethnicity, *n* (%)** - **Italian** - **Romanian** - **Moroccan** - **Asian** - **Other** - **Missing data**	843 (90.6) 34 (3.7) 13 (1.4) 1 (0.1) 36 (3.9) 3 (0.3)
**Migraine, MA/MO—*n* (%)**	230/700 (24.7/75.3)
**1B—Control Group**	
**Variables**	Patients (*n* = 300)
**Sex, M/F—*n* (%)** - **<12 years, M/F—*n* (%)** - **≥12 years, M/F—*n* (%)**	152/148 (50.7/49.3) 105/111 (48.6/51.4) 47/37 (55.9/44.1)
**Age at diagnosis—mean ± SD (years)**	8.5 ± 4.3
**Ethnicity, *n* (%)** - **Italian** - **Romanian** - **Moroccan** - **Asian** - **Other**	253 (84.3) 11 (3.7) 14 (4.7) 5 (1.7) 17 (5.7)

F: female; M: male; SD: standard deviation; MA: migraine with aura; MO: migraine without aura.

**Table 2 jcm-12-02582-t002:** Characteristics of the study population after stratification into five age groups (≤5 years, 6–8 years, 9–11 years, 12–14 years, and 15–18 years).

Age Group at Diagnosis (years)	Patients *n* (%)	F *n* (%)	MA *n* (%)	F with MA *n* (%)	M with MA *n* (%)	MO *n* (%)	F with MO *n* (%)	M with MO *n* (%)
≤5	38 (4.1)	13 (34.2)	1 (2.6)	0 (0.0)	1 (100)	37 (97.4)	13 (35.1)	24 (64.9)
6–8	173 (18.6)	82 (47.4)	13 (7.5)	7 (53.8)	6 (46.2)	160 (92.5)	75 (46.9)	85 (53.1)
9–11	401 (43.1)	177 (44.1)	73 (18.2)	43 (58.9)	30 (41.1)	328 (81.8)	134 (40.9)	194 (59.1)
12–14	281 (30.2)	149 (53.0)	117 (41.6)	74 (63.3)	43 (36.7)	164 (58.4)	75 (45.7)	89 (54.3)
15–18	37 (4)	25 (67.6)	26 (70.3)	17 (65.4)	9 (34.6)	11 (29.7)	8 (72.7)	3 (27.3)
Total	930 (100)	446 (48.0)	230 (24.7)	141 (15.1)	89 (9.6)	700 (75.3)	305 (32.8)	395 (42.5)

F: female; M: male; MA: migraine with aura; MO: migraine without aura.

**Table 3 jcm-12-02582-t003:** Family history of major thrombotic events at ≤50 years old in children with MA and MO and children in the control group.

**Family History of Vascular Diseases ≤50 Years**		
MA (20.3%) vs. MO 89 (13.4%)	(*p* = 0.005)	OR =1.6; CI 1.1–2.4
MA (20.3%) vs. control group (14.7%)	(*p* = 0.02)	OR = 1.48; CI 0.93–2.34
**Family History of Stroke and TIA**		
MA (12.4%) vs. MO (5.1%)	(*p* < 0.001)	OR = 2.6; CI 1.54–4.46
MA (12.4%) vs. control group (4.3%)	(*p* < 0.001)	OR = 3.1; CI 1.6–6.2
**Family History of DVT**		
MA (4.6%) vs. MO (1.7%)	(*p* = 0.01)	OR = 2.86; CI 1.2–6.8

MA: migraine with aura; MO: migraine without aura; OR: odds ratio; CI: confidence interval; TIA: transitory ischemic attack; DVT: deep venous thrombosis.

## Data Availability

The data presented in this study are available on request from the corresponding author. The data are not publicly available due to our institution’s policy.

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
