# Peer review of "Cardiovascular Risk Factors and Family History of Major Thrombotic Events in Children with Migraine: A 12-Year Retrospective Single-Centre Study"

_jcm, 2023, doi:10.3390/jcm12072582_

Round 1

Reviewer 1 Report

Overall, it is an interesting and well written manuscript. 

Some comments to be addressed;

1. Please explain on the way that the family history was obtained. Are the family members were treated in the same hospital as well and their documents were accessible? How do you justifify the accurateness of the family history?

2. Besides MA and MO classification, why did not you analyse other types of migraine as well? Esp hemiplegic migraine. 

3. Please justify the current methodology used, and why case control method was not used. Also, please do a comparison between all migraine patients vs control. (Current result comparing the subanalysis MA/MO/Control is quite confusing)

4. Please add in the discussion potential genes that may contribute to your study findings.

5. Some minor editing need to be done at line 163 and line 174, the sentence ''Primary headaches are often related to a family history of cardiovascular dis- 174 eases. [17]'' was hanging without further explanation

Author Response

We have found every suggestion very useful to improve our article. Due to the Editor’s request, we have revised the article following the reviewers’ suggestion. The paper was revised by an English native speaker, as requested by Reviewer 2.

Some comments to be addressed;

  1. Please explain on the way that the family history was obtained. Are the family members were treated in the same hospital as well and their documents were accessible? How do you justify the accurateness of the family history?

Thank you for your question. Family history was obtained from interviews to parents and from written medical reports (e.g. MRI, blood tests) provided by parents. We have clearly stated how we have collected family history in the Methods section (page 4, lines 22-23).

  1. Besides MA and MO classification, why did not you analyse other types of migraine as well? Esp hemiplegic migraine. 

Thank you for your question. As usually done in studies on pediatric migraine, we have decided only to distinguish MA and MO, and not to analyse subtypes of MA (such as hemiplegic migraine) because children often show different combination of auras instead that single features (visual, sensorial, motor,…). In particular, in our population of children with MA, 179 showed visual aura (77.8%), 76 sensorial aura (33.0%), 44 motor aura (19.1%), and 43 showed dysarthria (18.7%); overall, 91 children (39.6%) showed two or more aura features on each episode.

Moreover, as we have already shown in a previous study on children with migraine (Rossi R, et al. Cephalalgia 2016), that acute hypotonia of an arm or, less frequently, a leg is extreme rare and should more properly described as “weakness” than true paresis, which is otherwise typical of hemiplegic migraine.

We have added the full description of aura features in our MA population in the results (page 6, lines 18-20).

  1. Please justify the current methodology used, and why case control method was not used. Also, please do a comparison between all migraine patients vs control. (Current result comparing the subanalysis MA/MO/Control is quite confusing)

Actually, ours was a retrospective case-control observation study, as now clearly stated in the Methods section (page 4, line 2).

As regards the comparison between all migraine patients and controls, no significant differences emerged, probably because only MA show higher risk of family history for cardiovascular events. For these reasons, we have decided not to include such comparison in our paper.

  1. Please add in the discussion potential genes that may contribute to your study findings.

Under your suggestion, we have improved the discussion on genetic factors potentially involved in the pathogenesis of MA and ischemic events (page 8, lines 24-26; page 9, lines 1-10).

  1. Some minor editing need to be done at line 163 and line 174, the sentence ''Primary headaches are often related to a family history of cardiovascular dis- 174 eases. [17]'' was hanging without further explanation

Due to your proper observation, we have moved the reported sentence to page 8, lines 4-5.

Reviewer 2 Report

Interesting and important paper.

I suggest to elaborate more on the underlying potential link between migraine and stroke (e.g. cortical spreading depression). Please discuss within this context also the phenomenon of hemiplegic migraine e.g. cite Saleh C, Pierquin G, Beyenburg S. Hemiplegic Migraine Presenting with Prolonged Somnolence: A Case Report. Case Rep Neurol. 2016 Oct 3;8(3):204-210 and discuss more the association between migraine and stroke and the similarity in imaging, see and cite e.g. Zhang YParikh AQianMigraine and stroke

Author Response

We have found every suggestion very useful to improve our article. Due to the Editor’s request, we have revised the article following the reviewers’ suggestion. The paper was revised by an English native speaker, as requested by Reviewer 2.

I suggest to elaborate more on the underlying potential link between migraine and stroke (e.g. cortical spreading depression). Please discuss within this context also the phenomenon of hemiplegic migraine e.g. cite Saleh C, Pierquin G, Beyenburg S. Hemiplegic Migraine Presenting with Prolonged Somnolence: A Case Report. Case Rep Neurol. 2016 Oct 3;8(3):204-210 and discuss more the association between migraine and stroke and the similarity in imaging, see and cite e.g. Zhang Y, Parikh A, Qian S Migraine and stroke Stroke and Vascular Neurology 2017;2:doi: 10.1136/svn-2017-000077

Thank you for your suggestion. We have implemented the discussion section and the references as requested (page 8, lines 8-18 and 24-26; page 9, lines 1-10).

Please provide in bullet form take home points apart from the conclusion section.

We have provided take home messages in bullet form after the conclusions (page 11 and 12), as follows:

  • Children with migraine with aura showed had higher risk of showing family history for cerebral ischemic events and deep venous thrombosis
  • Detection of family history for early ischemic events and any personal inborn or acquired risk factor is mandatory in children with migraine as well as in adults
  • The identification of effective biomarkers to stratify the risk of major thrombotic events in patients with migraine with aura is desirable

Round 2

Reviewer 2 Report

The authors have replied to all questions. The paper is improved. I have no further comments.